# Adipose-Derived Mesenchymal Stem Cells do not Affect the Invasion and Migration Potential of Oral Squamous Carcinoma Cells

**DOI:** 10.3390/ijms21186455

**Published:** 2020-09-04

**Authors:** Snehadri Sinha, Matilda Narjus-Sterba, Katja Tuomainen, Sippy Kaur, Riitta Seppänen-Kaijansinkko, Tuula Salo, Bettina Mannerström, Ahmed Al-Samadi

**Affiliations:** 1Department of Oral and Maxillofacial Diseases, Clinicum, University of Helsinki, 00014 Helsinki, Finland; matilda.narjus-sterba@helsinki.fi (M.N.-S.); katja.tuomainen@helsinki.fi (K.T.); sippy.kaur@helsinki.fi (S.K.); riitta.seppanen-kaijansinkko@helsinki.fi (R.S.-K.); tuula.salo@helsinki.fi (T.S.); bettina.mannerstrom@helsinki.fi (B.M.); ahmed.al-samadi@helsinki.fi (A.A.-S); 2Helsinki University Central Hospital, 00290 Helsinki, Finland; 3Translational Immunology Research Program (TRIMM), Faculty of Medicine, University of Helsinki, 00014 Helsinki, Finland; 4Cancer and Translational Medicine Research Unit, University of Oulu, 90014 Oulu, Finland; 5Medical Research Centre, Oulu University Hospital, 90014 Oulu, Finland

**Keywords:** mesenchymal stem cells, adipose tissue, bone marrow, oral squamous cell carcinoma, invasion, migration

## Abstract

Mesenchymal stem cells (MSCs) are commonly isolated from bone marrow and adipose tissue. Depending on the tissue of origin, MSCs have different characteristics and physiological effects. In various cancer studies, MSCs have been found to have either tumor-promoting or tumor-inhibiting action. This study investigated the effect of adipose tissue-MSCs (AT-MSCs) and bone marrow-MSCs (BM-MSCs) on global long interspersed nuclear element-1 (LINE-1) methylation, the expression level of microenvironment remodeling genes and cell proliferation, migration and invasion of oral tongue squamous cell carcinoma (OTSCC). Additionally, we studied the effect of human tongue squamous carcinoma (HSC-3)-conditioned media on LINE-1 methylation and the expression of microenvironment remodeling genes in AT-MSCs and BM-MSCs. Conditioned media from HSC-3 or MSCs did not affect LINE-1 methylation level in either cancer cells or MSCs, respectively. In HSC-3 cells, no effect of MSCs-conditioned media was detected on the expression of *ICAM1, ITGA3* or *MMP1*. On the other hand, HSC-3-conditioned media upregulated *ICAM1* and *MMP1* expression in both types of MSCs. Co-cultures of AT-MSCs with HSC-3 did not induce proliferation, migration or invasion of the cancer cells. In conclusion, AT-MSCs, unlike BM-MSCs, seem not to participate in oral cancer progression.

## 1. Introduction

Mesenchymal stem cells (MSCs) comprise a heterogeneous source of adherent cells that express specific surface markers, e.g., CD73, CD90 and CD105, and have the capacity for multilineage differentiation [1]. MSCs release various cytokines and growth factors that, through paracrine signaling, modulate a range of acute and chronic pathological conditions [2,3]. Though MSCs are present in all organs of the body, they are commonly isolated from the bone marrow or adipose tissue. Bone marrow-derived MSCs (BM-MSCs) were the first MSCs to be discovered and to date are also the most studied in terms of characterization and clinical applications [4]. However, adipose tissue-derived MSCs (AT-MSCs) offer the operational advantages of easier and safer isolation procedures, collection of larger volume of tissue material and higher proliferative capacity [1,5]. BM-MSCs and AT-MSCs have several differences in their characteristics, functions and clinical applications, despite sharing a few similarities [6,7]. 

In recent decades, the role of MSCs in cancer has generated interest and several studies have been carried out to that end, but the results have been controversial. Some studies attribute both AT-MSCs and BM-MSCs to promoting tumor progression and metastasis [6,8,9]. Cells co-expressing MSC markers CD73, CD90 and CD105 were reported to be enriched in head and neck squamous cell carcinoma [10]. MSC-derived factors act on cancer cells to release cytokines, inflammatory mediators and angiogenic factors, and simultaneously inhibit immune cell function [11]. However, there are also studies that indicate tumor-suppressing activities of MSCs. He and colleagues reported that conditioned media from umbilical cord MSCs inhibited breast cancer cell growth, migration and metastasis [12]. BM-MSCs were reported to inhibit angiogenesis and induce apoptosis [13,14]. Nevertheless, the immunosuppressive function of MSCs in general is not always observed, as it depends on variables such as the cell dose, proximity to immune cells, and stimulation by inflammatory cytokines [15]. Given that MSCs are often branded as a promising therapeutic tool, it is necessary to examine MSCs from different sources and in different contexts to understand their biological functions and evaluate their clinical prospects better.

Using a 3-dimensional (3D) myoma organotypic invasion model, we have previously investigated the interaction of BM-MSCs with oral tongue squamous cell carcinoma (OTSCC) cell lines, focusing on the analysis of invasion and gene expression profiles. Our results demonstrated that BM-MSCs were able to promote the invasion of OTSCC cells via inducing the expression of genes linked to chemokine signaling, epithelial plasticity, cell motility and invasion [16].

In this study, we first analyzed the effect of conditioned media from AT-MSCs and BM-MSCs on global long interspersed element-1 (LINE-1) methylation levels in human tongue squamous carcinoma (HSC-3) cells, and the same for HSC-3-derived conditioned media on AT-MSCs and BM-MSCs (Figure 1). LINE-1 is a repetitive DNA element comprising nearly a fifth of the human genome and its methylation level may reflect global genomic methylation [17]. Global LINE-1 methylation has been observed to precede tumor development and may serve as an early indicator of cancer [18]. In parallel, we analyzed changes in expression of intercellular adhesion molecule 1 (*ICAM1*), integrin alpha-3 (*ITGA3*) and matrix metallopeptidase 1 (*MMP1*) expression in AT-MSCs, BM-MSCs and HSC-3 cells in this bidirectional study. ICAM1, ITGA3 and MMP1 are involved in the remodeling of the extracellular matrix, and their expression is known to be aberrant in some cancers [19,20,21]. We also examined the tumorigenic potential of AT-MSCs on OTSCC by evaluating the proliferation, migration and invasion of HSC-3 and dysplastic oral keratinocyte (DOK) cells.

## 2. Results

### 2.1. HSC-3- and MSCs-Conditioned Media had no effect on LINE-1 Methylation

In order to evaluate whether the methylation profile of HSC-3 cells could be affected by contents of the secretome (in conditioned media) from AT-MSCs and BM-MSCs, methylation analysis on a specific site of the LINE-1 element was performed. While the methylation-specific multiplex ligation-dependent probe amplification (MS-MLPA) assay used here includes three methylated sites in LINE-1, the result here was based on the site that is used in most LINE-1 methylation assays. HSC-3 cells were grown in conditioned media from AT-MSCs (*n* = 2) and BM-MSCs (*n* = 2) for four days, following which DNA was extracted for MS-MLPA. Average LINE-1 methylation in HSC-3 cells grown in both MSCs-conditioned media was marginally lower than that in HSC-3 cells grown in serum-free media (baseline, Figure 2A). Though conditioned media were taken from only two donors per MSC type, none of the conditioned media were indicated to have had any effect on LINE-1 methylation in HSC-3 cells. In the reverse setup, where AT-MSCs (*n* = 4) and BM-MSCs (*n* = 4) were grown in HSC-3-conditioned media for four days, the average LINE-1 methylation level in both MSC types grown in HSC-3-conditioned media was comparable to baseline LINE-1 methylation, based on cells grown in serum-free media (Figure 2B).

### 2.2. Microenvironment Remodeling genes Expression of HSC-3 and MSCs cells were Mainly Unaffected when Cultured in MSCs- or HSC-3-Conditioned Media

We then investigated whether conditioned media from AT-MSCs and BM-MSCs could affect the expression of genes involved in the microenvironment remodeling of HSC-3 cells. HSC-3 cells were grown in conditioned media from AT-MSCs (*n* = 2) and BM-MSCs (*n* = 2) for four days and evaluated for the expression of *ICAM1*, *ITGA3* and *MMP1* by quantitative real-time PCR (qRT-PCR) (Figure 3; Appendix A). *ICAM1* and *ITGA3* expression in HSC-3 cells growing in both types of MSCs-conditioned media was comparable to that of cells growing in serum-free media (control). *MMP1* expression was slightly higher in HSC-3 cells when grown in AT-MSC and BM-MSC media (around 1.5- and 2.5-fold, respectively) compared to the control.

In a similar way, AT-MSCs (*n* = 2) and BM-MSCs (*n* = 2) were grown in HSC-3-derived conditioned media for four days, and expression levels of the same genes were analyzed (Figure 3B; Appendix A). Here, only *ITGA3* expression was comparable to the control. *ICAM1* levels were 14- and 5-fold higher in AT-MSCs and BM-MSCs, respectively, whereas *MMP1* levels were higher by 6- to 60-fold in both cell types.

### 2.3. AT-MSCs had no Effect on the Invasion and the Proliferation Abilities of HSC-3 or DOK Cells

Since BM-MSCs were previously shown to induce oral cancer growth, we investigated if AT-MSCs have similar effects. We co-cultured HSC-3 and DOK cells with AT-MSCs and normal oral fibroblasts (NOFs) as the positive control using a 3D myoma disc organotypic model. Both the invasion area and depth were measured using ImageJ software based on pan-cytokeratin staining. Neither the invasion area nor the depth of invasion of HSC-3 and DOK cells were affected by adding AT-MSCs. On the other hand, NOFs, as a control, increased the invasion area and the depth of both HSC-3 and DOK. The increase in the invasion surface area reached statistical significance (Figure 4). Similarly, the number of the small tumor cell islands, budding, was increased when NOFs were added (Figure 4F,G).

To investigate if AT-MSCs have an effect on cancer cells’ proliferation, myoma samples were stained with Ki67 and the percentage of the positive cells was calculated. Neither AT-MSCs nor NOFs increased the proliferation ability of the cancer cells (Figure 5).

### 2.4. AT-MSCs and NOF did not Increase the Migration of HSC-3/DOK Cells

As cancer cells migration is an important feature for cancer progression, we used an IncuCyte wound healing migration assay to study the effect of AT-MSCs and NOFs on the cancer and dysplastic cells migration ability. HSC-3 and DOK were co-cultured with AT-MSCs and NOFs, and a wound was made using a WoundMaker™ tool. The wound closure rate was monitored for 24 h. Neither AT-MSCs nor NOFs increased, but rather decreased, the migration of HSC-3 and DOK cells (Figure 6).

## 3. Discussion

AT-MSCs and BM-MSCs share several biological features but also have a considerable number of differences altogether, therefore they are expected to differ in their functional effects. Initially, differences in their operational features, such as frequency in isolated tissue, ease of harvest, differentiation potential and proliferative capacity, may be considered for their use in a clinical context. Yet, differences in their behavior in tissue and in vivo environments require greater attention. Overall functions of MSCs include regulation of tissue homeostasis and support of tissue integrity. In general, MSCs have an innate tendency to migrate towards damaged tissue and wounds to promote regenerative activities [22,23].

With regard to cancer, it is known that tumor-derived cytokines and other soluble factors attract MSCs towards the tumor microenvironment, thereby playing a vital role in determining the tumor fate [24]. Tumor-associated MSCs have the ability to modulate the tumor microenvironment, since they can exert either stimulatory or inhibitory effects on tumor pathology [6]. For example, conditioned media from AT-MSC reduced the metabolic activity and viability of renal and bladder carcinoma, but also increased the cancer cells’ resistance to chemotherapy [25]. Effects of MSCs are also context-dependent: one study reported anti-tumor effects of BM-MSCs on lung cancer and esophageal cancer in vitro, yet they were found to enhance tumor growth in vivo in the same study [26]. With current evidence suggesting that AT-MSCs and BM-MSCs show both anti- and pro-tumor effects in various cancers, this dual function of MSCs could be influenced by several factors, such as: MSCs origin and source, cancer type and the secretome of MSCs and its interaction with the host immune system and other cells in the tumor microenvironment.

Our findings showed that conditioned media from both AT-MSCs and BM-MSCs had very subtle effects on the LINE-1 methylation of HSC-3 cells, and the same when AT-MSCs and BM-MSCs were grown in conditioned media derived from HSC-3. Hypomethylation of LINE-1 is a common phenomenon in many cancers [27], and was also observed in one of our recent studies where AT-MSCs were treated with osteosarcoma-derived extracellular vesicles (EVs) [28]. Those EVs were found to influence AT-MSCs at both the genetic and epigenetic level. In this study, however, no significant changes in methylation level were observed with the conditioned media.

The conditioned media from HSC-3 cells induced upregulation of ICAM1 and MMP1 in both AT- and BM- MSCs. *ICAM1* expression was upregulated to a higher degree in AT-MSCs than in BM-MSCs, while *MMP1* had comparable levels of upregulation in both AT- and BM-MSCs. ICAM1 facilitates intercellular interactions between cells and is implicated in promoting the invasiveness and metastatic ability of cancers [29,30]. Meanwhile, MMP1 has been shown to be necessary for the migration of BM-MSCs toward gliomas [31]. MMP1 and other matrix metalloproteinases (MMPs) are upregulated in OTSCC and promote invasion [32,33]. Lastly, no changes in *ITGA3* expression were observed in any of the cells, though it has been reported to be over-expressed in many cancers [20]. Though normal MSCs did not affect gene expression or LINE-1 methylation in HSC-3 cells, it is possible that MSCs conditioned with HSC-3 media may have pro-tumor effects on HSC-3 cells.

Proliferation, migration, invasion and budding formation are important features needed for cancer progression. AT-MSCs have previously been reported to promote proliferation, invasion and MMPs secretion in epithelial ovarian cancer [34]. Additionally, we have reported that BM-MSCs reduced the proliferation and enhanced the invasion of OTSCC [16]. In this study, however, we were not able to detect similar effects of AT-MSCs on HSC-3 and DOK cells. Conditioned media from AT-MSCs also had minimal effects on *MMP1* expression in HSC-3 cells. This clearly shows the differences in the behavior of MSCs based on their origin and the target cells. These results point out that in contrast to BM-MSCs, AT-MSCs do not enhance the aggressiveness of oral dysplastic and cancer cells in vitro.

MSCs offer much promise in regenerative medicine, yet the controversies surrounding the functional effects in vivo and safety concerns hinder their widespread approval. At the moment, AT-MSCs are not as popular as BM-MSCs for therapeutic applications, but their relatively easier harvesting procedures and greater yield from tissues are attractive features that have contributed to a growing interest in AT-MSCs. Nevertheless, functional and clinical studies are of essence to determine the safety and efficiency of AT-MSCs, before they can be established as a viable alternative to BM-MSCs. Furthermore, donor-linked features introduce heterogeneities in MSC behavior, therefore quality control assessments would be necessary to deem them fit-for-purpose. This study indicated that AT-MSCs do not enhance the invasive or migratory potential of OTSCC cells, nor do they influence the expression of microenvironment-remodeling genes in OTSCC. Hence, AT-MSCs are suggested to be more benign than BM-MSCs against OTSCC, with previous findings in consideration. However, this study consisted of a limited number of samples and in vitro investigations, with only OTSCC cells as a target. Analyses of other functional effects and long-term monitoring of in vivo studies are required before making a case for the use of AT-MSCs in clinical applications.

## 4. Materials and Methods

### 4.1. Cell Cultures

Human AT-MSCs were derived from surplus adipose tissue received from plastic surgery procedures at the Department of Plastic Surgery (Laser Tilkka Ltd., Helsinki, Finland). The study was carried out under approval of the ethical committee of Helsinki and Uusimaa Hospital District and with informed consent from the donors (Ethical approval DNro: 217/13/03/02/2015 and HUS/3061/2019). AT-MSCs were obtained from liposuction aspirates from donors using mechanical and enzymatic isolation as described previously [35]. All donors were female, age range 37–54 years, average 46.4 years; body mass index (BMI) range 24–32 kg/m^2^, average 26.7 kg/m^2^. Cells were cultured in AT-MSCs media consisting of Dulbecco’s modified Eagle’s medium (DMEM)/Ham’s Nutrient Mixture F-12 with 1% L-alanyl-L-glutamine (DMEM/F-12 1:1 GlutaMAX; Gibco, Paisley, UK), 1% antibiotics (100 U/mL penicillin, 0.1 mg/mL streptomycin; Lonza, Verviers, Belgium) and 10% Fetal Bovine Serum (FBS, South American, Gibco). Cells were grown at +37 °C with 5% CO_2_ and under humid conditions. In this study, AT-MSCs were in passage 2–6. Once AT-MSCs adhered to the culture flask surface, non-adherent populations were washed away with PBS, and fresh media were added.

Human BM-MSCs were received as a kind gift from Assoc. Prof. Susanna Miettinen (University of Tampere, Tampere, Finland). Donors consisted of three male and one female; age range 69–92, average 84.5; BMI range 19–29, average 24.7. These cells were cultured similarly to AT-MSCs. Cells were in passage 3–4.

HSC-3 cells (Japan Health Sciences Foundation, Tokyo, Japan) cells and DOK (Sigma-Aldrich, St. Louis, MO, USA) were cultured using DMEM/F-12 (Gibco) supplied with 10% heat-inactivated FBS, 100 U/mL penicillin, 100 μg/mL streptomycin, 50 μg/mL ascorbic acid and 250 ng/mL fungizone (all from Sigma-Aldrich). NOFs were established from tissue explants as described previously [36] and cultured using DMEM supplemented with 10% FBS, 100 U/mL penicillin, 100 μg/mL streptomycin, 250 ng/mL fungizone, 1% sodium pyruvate (Sigma-Aldrich) and 50 μg/mL ascorbic acid.

### 4.2. MSCs Characterization

The MSCs were characterized according to the International Society Cell & Gene Therapy (ISCT) guidelines, i.e., plastic-adherence, specific surface marker expression and multipotent differentiation potential [37]. The multipotentiality of these cells has been reported in our previous papers [38,39]. AT-MSCs (*n* = 5) and BM-MSCs (*n* = 4) were characterized using a BD Accuri C6 flow cytometer (Becton Dickinson, Franklin Lakes, NJ, USA) to confirm the mesenchymal origin of the cells (Table 1). Allophycocyanin-conjugated monoclonal antibodies against CD14, CD19, CD34, CD45RO, CD54, CD73, CD90, CD105 and HLA-DR (Becton Dickinson) were used to stain at least 100,000 cells per antibody. Analysis was performed for 10,000 events with positive expression defined as the level of fluorescence greater than 99% of the corresponding unstained cell sample.

All nine MSC cell lines were positive for surface markers CD73, CD90 and CD105 in > 90% of their population, the only exceptions being one AT-MSC cell line with a lower expression of CD105 and one BM-MSC cell line with a slightly lower expression of CD90 (Table 1). MSCs were adherent and displayed multipotent differentiation. AT-MSCs had higher expression of CD34 and CD54 than BM-MSCs. Otherwise, both AT- and BM-MSCs had very low expression of CD14, CD19, CD45 and HLA-DR.

### 4.3. Culture of MSCs and HSC-3 Cells in Conditioned Media

A total of 500,000 cells of each cell type (AT-MSC, BM-MSC and HSC-3) were plated in T-175 flasks (Corning Inc., Corning, NY, USA) and grown until they approached 70% confluency. Then, the growth media were changed to serum-free conditions (no FBS) and collected after 24 h of culture. The confluency approach was taken to normalize the metabolite content in the conditioned media. Conditioned media were centrifuged at 600× *g*, after which the supernatant was passed through a 0.45 µm filter and stored at −80 °C until use. HSC-3 cells were grown for 24 h, then washed with PBS and grown in (AT- and BM-)MSCs-derived conditioned media for four days. AT-MSC and BM-MSC cells were grown for 24 h and let to adhere, then washed with PBS and grown in HSC-3-derived conditioned media for four days. In all cases, the same cells were grown in serum-free media as a control.

### 4.4. Gene Expression Analysis with qRT-PCR

Total RNA was extracted from AT-MSCs, BM-MSCs and HSC-3 cells using the NucleoSpin^®^ RNA isolation kit (Macherey-Nagel, Düren, Germany) and reverse-transcribed to cDNA using SuperScript™ IV VILO™ Master Mix (Thermo Fisher Scientific Baltics UAB, Vilnius, Lithuania). All qRT-PCR reactions were conducted in triplicates on MicroAmp™ Optical 96-well reaction plates using TaqMan assays, with reactions carried out in QuantStudio 5 (all from Life Technologies, Carlsbad, CA, USA). The expression of the following genes was quantified: intercellular adhesion molecule 1 (*ICAM1*, assay ID Hs00164932_m1), integrin alpha-3 (*ITGA3*, assay ID Hs01076879_m1) and matrix metallopeptidase 1 (*MMP1*, assay ID Hs00899658_m1). Expression levels were normalized with the housekeeping gene, ribosomal protein lateral stalk subunit P0 (*RPLP0*, assay ID Hs99999902_m1), and calculated using the ddCt-method [40].

### 4.5. LINE-1 Methylation Analysis

DNA was extracted from AT-MSCs, BM-MSCs and HSC-3 cells using NucleoSpin^®^ Tissue (Macherey-Nagel). For two samples that had a low cell number, NucleoSpin^®^ Tissue XS (Macherey-Nagel) was used. A total of 90 ng of DNA was used to perform the MS-MLPA reactions, using the SALSA MS-MLPA Reagents kit and P300-B1 Human DNA Reference-2 (MRC-Holland, Amsterdam, The Netherlands). A customized set of LINE-1 probes was used according to a previously established protocol [41,42]. Two controls were used: one commercially available colon cancer cell line RKO (ATCC^®^ CRL-2577™, Manassas, VA, USA) and one unmethylated sample generated using the GenomePlex complete whole genome amplification kit. The methylation dosage ratio was calculated as described previously [41].

### 4.6. Organotypic Myoma Discs

In order to study the effect of AT-MSCs on oral cancer (HSC-3) and dysplastic (DOK) cell proliferation and invasion, we have developed an in vitro model using human uterine leiomyoma discs [43]. NOFs were used as the positive control. The use of myoma tissue was approved by the Ethics Committee of Oulu University Hospital, and all patients signed an informed consent form. We prepared uterine myoma discs and completed invasion experiments with some modifications, as described previously [43,44]. Briefly, we placed myoma discs into Transwell^®^ inserts (Corning Inc., Corning, NY, USA) and 40 × 10^4^ HSC-3 or DOK with or without 20 × 10^4^ AT-MSCs or NOFs seeded on top of the discs. The cells were allowed to adhere overnight, and on the next day, the discs were transferred onto a steel grid in a 12-well plate supplied with 1 mL culture media replaced every 3 to 4 days for 10 days. Myoma discs were fixed in 10% formalin and embedded in paraffin blocks.

### 4.7. Immunohistochemical Staining of the Myoma Discs

Samples were cut into 4-μm-thick sections and stained using a fully automated Leica BOND-MAX staining robot (Leica Microsystems, Wetzlar, Germany) and a horseradish peroxidase–labeled dextran polymer method. As for primary antibodies, we used monoclonal mouse anti-human pan-cytokeratin (0.7 μg/mL) and Ki67 (0.8 μg/mL), both from Dako (Dako, Glostrup, Denmark).

### 4.8. Scratch-Wound Cell Migration Assay

To test the effect of AT-MSCs and NOFs on HSC-3 and DOK cell migration, wells of a 96-well Imagelock plate (Essen Bioscience, Ann Arbor, MI, USA) were coated with 300 μg/mL human Myogel matrix [45] and incubated overnight (37 °C). Leftover Myogel was removed by suction and cell suspensions (100 µL/well) were seeded and left to adhere overnight in an incubator. Cell densities were optimized to achieve a confluent cell layer after 24 h; HSC-3 (25,000 cells/well), NOFs (12,500 cells/well), AT-MSCs (12,500 cells/well) and DOK (20,000 cells/well). Cell lines were seeded alone and as a mixture of two cell lines. For preparation of homogeneous wounds, we used a WoundMaker™ tool (Essen Bioscience). After wound making, cell culture media were replaced with 100 μL of fresh media. Wound confluence was monitored using the IncuCyte Zoom Live-Cell Imaging System (Essen Bioscience) by taking images every second hour for 24 h.

### 4.9. Microscopic and Histomorphometric Analysis

Slides were imaged using a Leica DM6000 B/M light microscope connected to a digital camera (DFC420; Leica Microsystems). We used ImageJ 1.51 (64-bit) software to measure the invasion area, the invasion depth and the average cell island size, budding, as described previously [43,44,46]. The percentage of Ki67^+^ cells was calculated from at least three randomly selected fields of the non-invading cells (on the myoma surface).

### 4.10. Statistical Analysis

When characterizing MSCs from different donors in terms of surface marker expression, values were reported as means ± standard deviations. Gene expression values were also provided as means ± standard deviations. Myoma invasion experiments were repeated four times independently, each in duplicate. Values are given as means ± standard deviations. To determine the statistical significance, we performed one-way analysis of variance (ANOVA) followed by the Bonferroni post hoc test. We set statistical significance to *P* < 0.05.

## Figures and Tables

**Figure 1 ijms-21-06455-f001:**
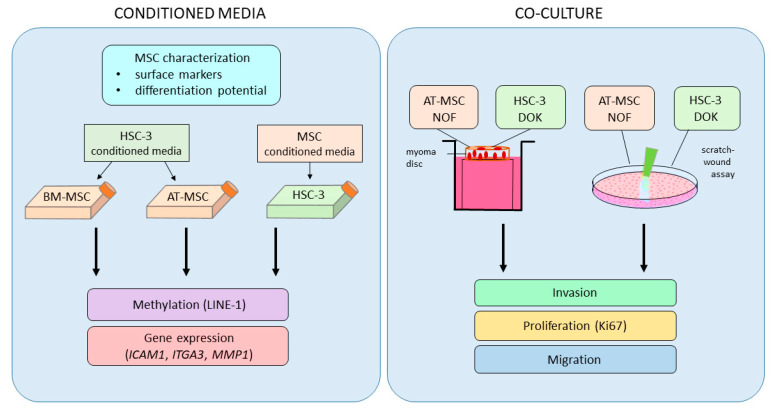
Study design workflow. Adipose tissue-derived mesenchymal stem cells (AT-MSCs) and bone marrow-derived mesenchymal stem cells (BM-MSCs) were first characterized and then used for conditioned media studies with human tongue squamous carcinoma (HSC-3) cells. Cells growing in conditioned media were analyzed for long interspersed nuclear element-1 (LINE-1) methylation and changes in expression of microenvironment remodeling genes. AT-MSCs, HSC-3 cells, dysplastic oral keratinocyte (DOK) cells, and normal oral fibroblasts (NOFs) were used in a co-culture system for invasion, proliferation and migration studies.

**Figure 2 ijms-21-06455-f002:**
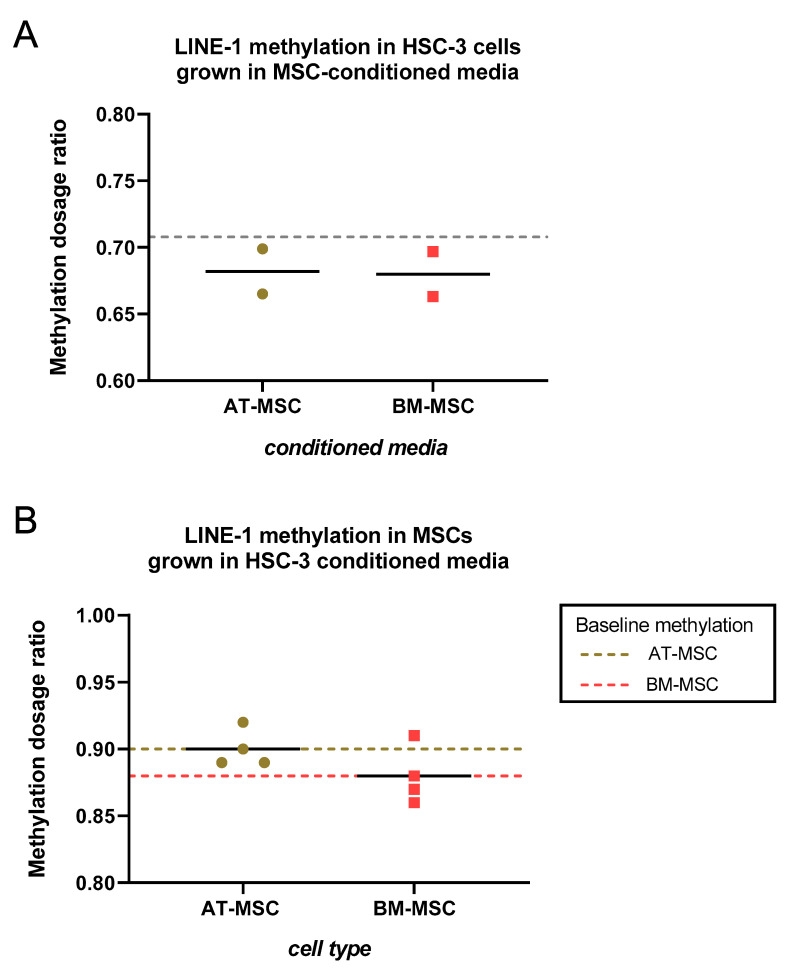
Effect of HSC-3- and mesenchymal stem cells (MSCs)-conditioned media on LINE-1 methylation in target cells. (**A**) HSC-3 cells were grown in (AT- and BM-) MSCs-derived conditioned media. DNA was isolated from cells and LINE-1 methylation was analyzed by methylation-specific multiplex ligation-dependent probe amplification (MS-MLPA). The dashed line, baseline methylation, is based on HSC-3 cells grown in serum-free media. (**B**) AT-MSCs (olive green) and BM-MSCs (red) were grown in HSC-3-derived conditioned media. DNA was isolated from cells and LINE-1 methylation was analyzed by MS-MLPA. Dashed lines represent baseline methylation for each cell type, based on MSCs growing in serum-free media.

**Figure 3 ijms-21-06455-f003:**
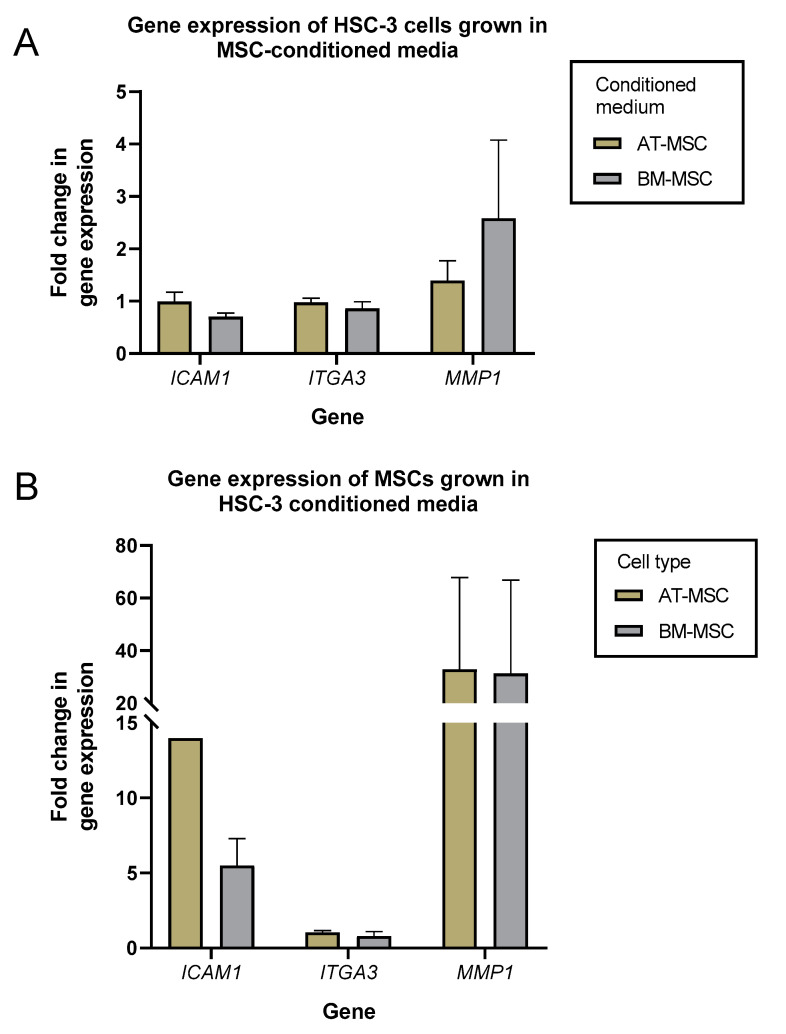
*ICAM1*, *ITGA3* and *MMP1* gene expression of HSC-3 and MSCs cells were mainly unaffected when cultured in MSCs- or HSC-3-conditioned media. HSC-3 cells were grown in (AT- and BM-) MSCs-derived conditioned media (**A**), while AT-MSCs and BM-MSCs were grown in HSC-3-derived conditioned media (**B**). RNA was isolated from cells and reverse-transcribed to cDNA, then analyzed for changes in gene expression of *ICAM1*, *ITGA3* and *MMP1*. Fold-change values were normalized to expression levels of the same cells grown in serum-free media.

**Figure 4 ijms-21-06455-f004:**
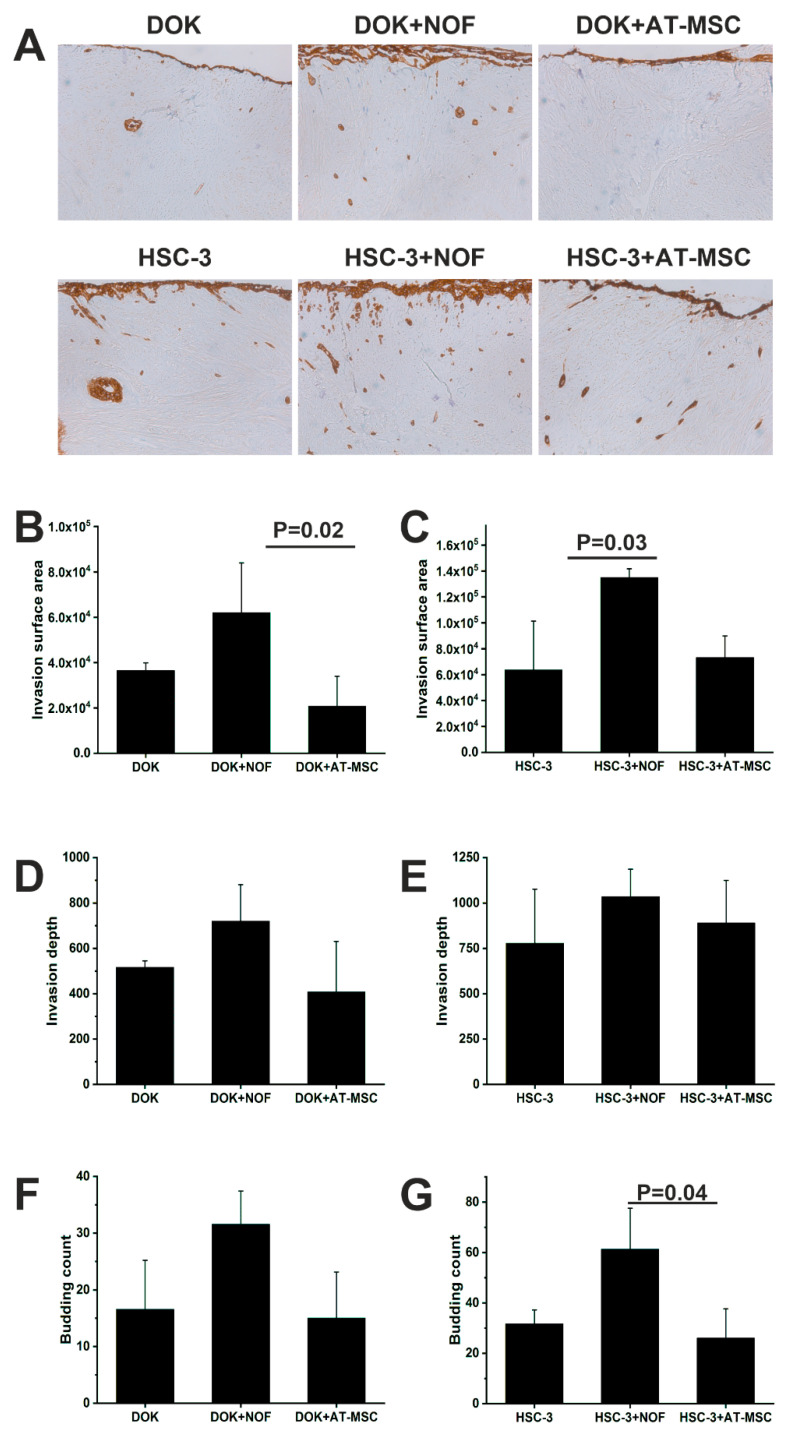
Invasion of DOK and HSC-3 cells was not affected by the presence of AT-MSCs. (**A**) Representative pictures of the pan-cytokeratin-stained cells in the myoma discs, pictures were taken at magnification 4×. DOK (**B,D,F**) and HSC-3 (**C,E,G**) cells were co-cultured with AT-MSCs and NOF using myoma organotypic model. Cells’ invasion and their ability to form budding were studied.

**Figure 5 ijms-21-06455-f005:**
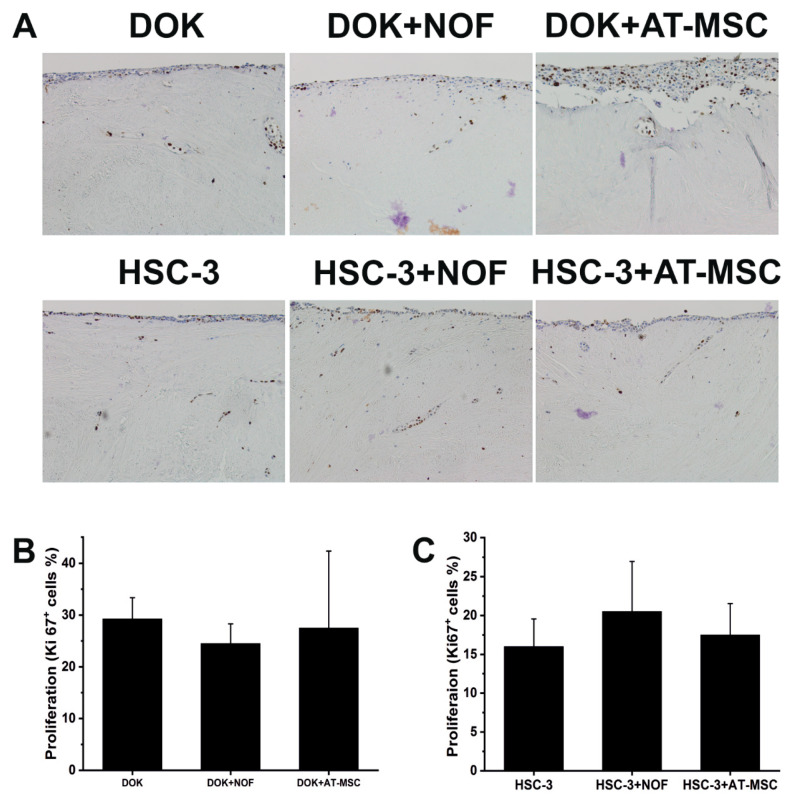
Proliferation of DOK and HSC-3 was not affected by the presence of AT-MSCs. (**A**) DOK and HSC-3 cells were co-cultured with AT-MSCs and NOF using a myoma organotypic model, pictures were taken at magnification 10×. (**B,C**) Ki67 was used as a marker for cell proliferation.

**Figure 6 ijms-21-06455-f006:**
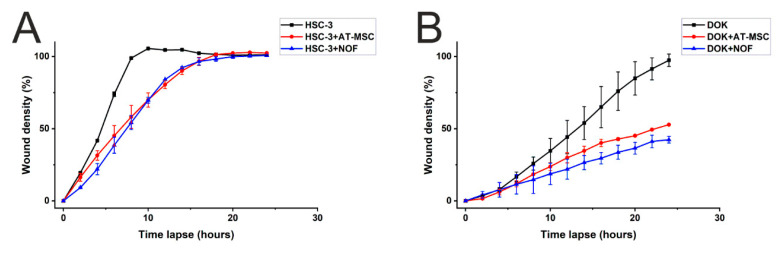
AT-MSC and NOF did not increase the migration of HSC-3 and DOK cells. IncuCyte wound migration assay was conducted to study the effect of AT-MSC and NOF on the migration of (**A**) HSC-3 cells and (**B**) DOK cells.

**Table 1 ijms-21-06455-t001:** Characterization of AT-MSCs and BM-MSCs by surface marker expression with flow cytometry analysis. Clone information for marker antibodies is also provided.

Marker	Clone	% Expression in Cells (Average ± Standard Deviation)	
		AT-MSC (*n* = 5)	BM-MSC (*n* = 4)
CD14	M5E2	1.7 ± 2.8	0.4 ± 0.3
CD19	HIB19	0.3 ± 0.4	0.3 ± 0.2
CD34	581	31.9 ± 25.4	1.1 ± 1.4
CD45RO	UCHL1	1.4 ± 2.3	0.5 ± 0.3
CD54	HA58	61.8 ± 22.8	12.4 ± 2.2
CD73	AD2	99.0 ± 1.8	99.5 ± 0.3
CD90	5E10	99.3 ± 0.7	94.6 ± 5.1
CD105	266	96.1 ± 7.1	98.2 ± 1.0
HLA-DR	G46-6	1.7 ± 3.1	1.5 ± 0.8

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
