# Peer review of "Adipose-Derived Mesenchymal Stem Cells do not Affect the Invasion and Migration Potential of Oral Squamous Carcinoma Cells"

_ijms, 2020, doi:10.3390/ijms21186455_

Round 1

Reviewer 1 Report

This is an interesting and well written paper.

I just suggest to improve the discussion introducing the clinical relevance supported by these results.

Author Response

Point 1: Improve the discussion introducing the clinical relevance supported by these results.

Response 1: We agree that the clinical significance of the study is relevant, so we have added at the end of the discussion accordingly (lines 205-219). For your convenience, the paragraph pertaining to the clinical relevance is as follows:

MSCs offer much promise in regenerative medicine, yet the controversies surrounding the functional effects in vivo and safety concerns hinder their widespread approval. At the moment, AT-MSCs are not as popular as BM-MSCs for therapeutic applications, but their relatively easier harvesting procedures and greater yield from tissues are attractive features that have contributed to a growing interest in AT-MSCs. Nevertheless, functional and clinical studies are of essence to determine the safety and efficiency of AT-MSCs, before they can be established as a viable alternative to BM-MSCs. Furthermore, donor-linked features introduce heterogeneities in MSC behavior, therefore quality control assessments would be necessary to deem them fit-for-purpose. This study indicated that AT-MSCs do not enhance the invasive or migratory potential of OTSCC cells, nor do they influence the expression of microenvironment-remodeling genes in OTSCC. Hence, AT-MSCs are suggested to be more benign than BM-MSCs against OTSCC, with previous findings in consideration. However, this study consisted of a limited number of samples and in vitro investigations, with only OTSCC cells as a target. Analyses of other functional effects and long-term monitoring of in vivo studies are required before making a case for the use of AT-MSCs in clinical applications.

Reviewer 2 Report

In your previous report (ref.16) coculture system was applied. In the manuscript the effect of conditioned media was evaluated. Both system, coculture with BM-MSCs or AD-MSCs and the addition of their coditioned media should be wvaluated.

Or the manuscript should be revised involving the topics.  

Author Response

Point 1: In your previous report (ref.16) coculture system was applied. In the manuscript the effect of conditioned media was evaluated. Both systems, coculture with BM-MSCs or AD-MSCs and the addition of their conditioned media should be evaluated. Or the manuscript should be revised involving the topics.

Response 1: We would like to mention that the previous report and the current study are based on two different projects. The previous report was based exclusively on BM-MSCs, while in the current study the focus is on AT-MSCs. In this manuscript, we are referring to the previous study to compare the findings of BM-MSCs with AT-MSCs for the invasion, proliferation and invasion experiments. Since conditioned media studies were not performed earlier, both MSC types have been used in the current study. We hope this explains the rationale for the study design.

Reviewer 3 Report

This study analyzed the effect of MSCs on tumor cells and vice versa. In detail, the authors investigated the influence of adipose tissue-dervied MSCs (AT-MSCs) and bone marrow-derived MSCs (BM-MSCs) on global long interspersed nuclear element-1 (LINE-1) methylation and gene expression of microenvironment remodeling genes in MSCs and OTCC cells. A further focus was the analysis of OTCC's cell proliferation, migration and invasion in the presence or absence of MSCs. In summary, the authors concluded that, in contrast to BM-MSCs, AT-MSCs do not participate in cancer progression.

The manuscript is well prepared and easy to understand. The following aspects should be addressed to improve the quality of the manuscript.

1) The rational for investigating LINE-1 methylation should be explained in more detail in the introduction and results section.

2) In figure 2, standard deviation is quite high in some conditions. Please include raw data to give readers a better impression of data distribution.

3) BM-MSC data should be presented in all figures as this paper is comparing the effect of AT-MSCs with BM-MSCs on cancer cell biology. The authors explain that part of these data have been published already. However, for a direct comparison it would be very beneficial to present a data set that directly compares both MSC types under the same conditions in one experiment.

4) Please extend the description of the experimental data in the results section. Particulary the second half of this section is very short. A more detailed description of the data will help the reader to understand the experiments, resulting conclusions and their importance.

5) As the authors compare supernatant from two types of MSCs it is neccessary to make sure that the same conditions have been used. Please comment on how many cells have been used and if conditions were comparable and how this was controlled. This is of particular interest as higher cell numbers in one condition (potentially due to different proliferation behavior) would result in more bioactive molecules in the medium and therefore in different effects on the target cell line.

Author Response

Point 1: The rational for investigating LINE-1 methylation should be explained in more detail in the introduction and results section.

Response 1: We have now provided the rationale for investigating LINE-1 methylation in the introduction (lines 66-69) with appropriate references, as follows:

LINE-1 is a repetitive DNA element comprising nearly a fifth of the human genome and its methylation level may reflect global genomic methylation. Global LINE-1 methylation has been observed to precede tumor development and may serve as an early indicator of cancer.

Additional details on LINE-1 methylation are also provided in the results section (lines 83-84).

Point 2: In figure 2, standard deviation is quite high in some conditions. Please include raw data to give readers a better impression of data distribution.

Response 2: We agree that the standard deviation is high in some cases, possibly caused by donor heterogeneity. The raw data from the experiments have been attached in Appendix A. The sample size in one instance has been corrected to 2 instead of 4 as earlier, after revisiting experimental notes.

Point 3: BM-MSC data should be presented in all figures as this paper is comparing the effect of AT-MSCs with BM-MSCs on cancer cell biology. The authors explain that part of these data have been published already. However, for a direct comparison it would be very beneficial to present a data set that directly compares both MSC types under the same conditions in one experiment.

Response 3: We would like to mention that the previous report and the current study are based on two different projects. The previous report was based exclusively on BM-MSCs, while in the current study the focus is on AT-MSCs. In this manuscript, we are referring to the previous study to compare the findings of BM-MSCs with AT-MSCs for the invasion, proliferation and invasion experiments. Since conditioned media studies were not performed earlier, both MSC types have been used in the current study. We hope this explains the rationale for the study design.

Point 4: Please extend the description of the experimental data in the results section. Particularly the second half of this section is very short. A more detailed description of the data will help the reader to understand the experiments, resulting conclusions and their importance.

Response 4: Description of data in the results section has now been elaborated upon, with the purpose for the assays mentioned. The findings have been explained further in the discussion section. The flowchart as Figure 1 has been included to help the reader understand the experiments.

Point 5: As the authors compare supernatant from two types of MSCs it is necessary to make sure that the same conditions have been used. Please comment on how many cells have been used and if conditions were comparable and how this was controlled. This is of particular interest as higher cell numbers in one condition (potentially due to different proliferation behavior) would result in more bioactive molecules in the medium and therefore in different effects on the target cell line.

Response 5: This is a valid point and it was important to us that the conditioned media from both MSC types was comparable in terms of metabolite content. The AT-MSCs and BM-MSCs had different doubling times, so we decided to normalize the conditioned media with respect to confluency of the flask. We seeded 500,000 cells for each MSC type in T175 flasks and let them grow until they approached 70% confluency. At that point, we changed the medium to serum-free conditions and collected the conditioned media after 24 hours of culture. We had alternatively planned to measure protein content in the media as a normalizing measure, but the phenol red in the media was an interfering agent with the bicinchoninic acid (BCA) assay. Moreover, protein absorbance with the Nanodrop spectrophotometer was not sensitive enough to measure protein concentration in the conditioned media. Therefore, equal volume of media was provided to each recipient cell flask. Some of this information has now been added to section 4.3 (in Materials and Methods) for readers to understand better.

Round 2

Reviewer 2 Report

Authors tried to evaluate the effects of MSCs on tumor cells or vise versa.

The need for evaluation of difference of MSCs origin on the effect on tumors is well described.

There's  few points to revise;

p3, lines 88-90,  The sample size, 2 is not enough to draw the conclusion " Neither ---- cells." At least 4 is needed  to clarify effect on methylation .

p3, lines 93, what is controls ?  not described in the Result  and not marked in the Figure. 

Author Response

Point 1: [p3, lines 88-90]:  The sample size, 2 is not enough to draw the conclusion “Neither ---- cells." At least 4 is needed to clarify effect on methylation.

Response 1: We agree that a sample size of two is not large enough to draw any conclusion. We have edited the text (lines 88-90) that there is an indication of neither cell type-derived conditioned media having had any effect on LINE-1 methylation. We have also pointed out that the methylation values are the average of both donors (line 86, line 91). The corrected sentence is as follows:

Though conditioned media was taken from only two donors per MSC type, none of the conditioned media were indicated to have had any effect on LINE-1 methylation in HSC-3 cells.

Point 2:  [p3, lines 93]: what is controls ?  not described in the Result and not marked in the Figure.

Response 2: By controls we meant baseline methylation, referring to the level of LINE-1 methylation of MSCs grown in serum-free media.  We have now made the sentence clearer (lines 91-93), and the edited version is such: 

In the reverse setup, where AT-MSCs (n=4) and BM-MSCs (n=4) were grown in HSC-3 conditioned media for four days, the average LINE-1 methylation level in both MSC types grown in HSC-3 conditioned media was comparable to baseline LINE-1 methylation, based on cells grown in serum-free media (Figure 2B).

Reviewer 3 Report

The authors have addressed all comments.

Author Response

Thank you for accepting our revision based on your constructive comments.